# An Update on Familial Mediterranean Fever

**DOI:** 10.3390/ijms24119584

**Published:** 2023-05-31

**Authors:** Maddalena Lancieri, Marta Bustaffa, Serena Palmeri, Ignazia Prigione, Federica Penco, Riccardo Papa, Stefano Volpi, Roberta Caorsi, Marco Gattorno

**Affiliations:** UOC Malattie Autoinfiammatorie e Immunodeficenze, IRCCS Istituto Giannina Gaslini, 16147 Genova, Italy

**Keywords:** Familial Mediterranean Fever, colchicine, colchicine resistance, children, autoinflammatory disease, Interlukin-1 Inhibitors

## Abstract

(1) Background: Familial Mediterranean Fever (FMF) is the prototypal autoinflammatory disease, characterized by recurrent bursts of neutrophilic inflammation. (2) Methods: In this study we look at the most recent literature on this condition and integrate it with novel information on treatment resistance and compliance. (3) Results: The canonical clinical presentation of FMF is in children with self-limited episodes of fever and polyserositis, associated with severe long-term complications, such as renal amyloidosis. It has been described anecdotally since ancient times, however only recently it has been characterized more accurately. We propose an updated overview on the main aspects of pathophysiology, genetics, diagnosis and treatment of this intriguing disease. (4) Conclusions: Overall, this review presents the all the main aspects, including real life outcome of the latest recommendation on treatment resistance of FMF, a disease, that not only helped understanding the pathophysiology of the auto inflammatory process but also the functioning of the innate immune system itself.

## 1. The Historical Background

Familial Mediterranean Fever (FMF) is the oldest and the most frequent autoinflammatory disease. It is a hereditary periodic fever syndrome characterized by self-limited episodes of fever and polyserositis associated with severe long-term complications, such as renal amyloidosis. FMF was particularly frequent in populations originating from the Mediterranean basin, such as Turks, Armenians, Jews, and Arabs [1]. The MEFV gene (from MEditerranean FeVer), located on the short arm of chromosome 16, was described for the first time in 1997 by the International and the French Consortium [2]. The protein encoded by the MEFV gene was initially called “Marenostrin” in reference to the Latin name of the Mediterranean Sea. Alternatively, the name “Pyrin” was given by the International Consortium in reference to the Greek name for fever. 

Before the identification of the causative gene, the description of the disease can be traced back to the ancient history of Mediterranean populations, in which the Galen were among the first to report it almost two thousand years ago. This condition continued to be part of Mediterranean history throughout the centuries despite migrations and the merging of different cultures and people. In 1945, Siegal defined “benign paroxysmal peritonitis” as an under-diagnosed and “unusual clinical syndrome” in himself and other patients: “The characteristics of this disorder are constant and distinctive. The syndrome is characterized by recurrent paroxysms of severe abdominal pain with fever which may be as high as 105° F [=~41.6 °C]. Chilliness or a shaking chill may accompany the attacks. Involvement of the peritoneum is indicated by the subjective symptom of marked abdominal soreness and the objective finding of widespread, exquisite direct and rebound tenderness” [3]. The name Familiar Mediterranean Fever was given in 1955 by Professor Heller and his study group [1], which became universal. 

In 1972, the first anecdotal observations on the efficacy of colchicine were provided by Goldfinger [4]. This event represented a revolution in the management of FMF, decreasing the frequency and intensity of the attacks and preventing renal amyloidosis, the most worrisome complication of uncontrolled FMF. Amyloid A (AA) amyloidosis results from continuous inflammation and unrestrained secretion of acute phase reactants. Colchicine was able to reduce the incidence of amyloidosis by reducing the levels of sub-clinical inflammation [5]. 

## 2. Epidemiology

FMF is prevalent in countries surrounding the Mediterranean Sea, especially affecting Turks, Arabs, Armenians, and Non-Ashkenazi Jews. Turkey is probably the country with the greatest prevalence, which is reported to be 1:1000 overall, with interregional differences. A nationwide multicentre study performed in Turkey [6] shows that patients with FMF originate mainly from the non-Mediterranean regions, with over 70% of the cases from central and eastern Anatolia and the inner Black Sea regions. Additional studies have revealed further differences in distribution, with the north-western region of Turkey having a prevalence as low as 6:10,000 [7]. Similarly, in Italy, the distribution of cases varies between northern and southern districts, the latter having a much higher occurrence of FMF. This phenomenon may be explained, at least partially, by the ancient colonization of the area by Greeks and Arabs and by the Jews’ migratory fluxes [8]. 

It is possible that MEFV mutations arose in pre-Biblical times and that Jews, being genetically isolated, might represent the most likely candidate founder population for several common MEFV mutations [9], with a prevalence in Israel of roughly 1–2:1000. In the Armenian population, the same prevalence has been calculated [10]. Additionally, the carrier rate of FMF mutations in Armenians was shown to be 1:5, as high as in North African and Iraqi Jews and Turks but lower than in Moroccan Jews (1:3.5) and Muslim Arabs (1:4.3). Such an elevated number of carriers, resulting from a founder effect, does not correlate with the real prevalence of patients with a diagnosis of FMF, since the detection of a single mutation (heterozygosity) does not help in making the diagnosis [11].

It has also been hypothesised in the past that the high carrier rate of the MEFV gene mutations in certain populations is the result of an evolutionary advantage against tuberculosis [12] or brucellosis [13]. The recent insights on the role of the pyrin inflammasome as a crucial sensor against infection from microbes producing exotoxins outlined the possible selective advantage of MEFV carriers towards the infection of *Yersinia pestis* during different devastating plagues hitting the Mediterranean basin during the centuries [14].

In addition to the above countries, FMF is found in North African countries, Greece, Crete, France, Germany, and the US. In most of these countries, the presence of FMF is largely related to robust emigration from Mediterranean countries. Many studies have shown the presence of different severity in FMF according to the country of residence, totally or partially independent of the pathogenicity of MEFV variants and ethnicity. The incidence of amyloidosis is much higher in Turk and Armenian patients living in their country of origin with respect to the same population emigrating to northern Europe or the US [15]. The same phenomenon was also reported in children by the international Eurofever registry, which showed how children living in Western Europe displayed a less severe disease activity independently from their ethnicity [16]. These observations likely reflect the burden of environmental factors (i.e., infections) as possible triggers for a more robust inflammatory response in Mediterranean countries.

Finally, a milder form of FMF is also present in Japan, with a lower prevalence of abdominal manifestations, a higher median age of onset, and a lower frequency of complications (AA amyloidosis) as compared to Mediterranean patients, probably due to differences in MEFV gene mutations [17].

## 3. Pathogenesis

Pyrin, the protein product of the MEFV gene, is an immunoregulatory molecule made up of 781 amino acids, interacting with the inflammasome components that can be activated in response to microbes. The protein is mainly expressed in granulocytes and dendritic cells and within serosal and synovial fibroblasts [18]. 

Pyrin contains an N-terminal eponymous PYD domain, central B-box zinc finger, bZIP transcription factor, coiled-coil domains, and a C-terminal B30.2 domain (Figure 1). Most FMF causative mutations are found in the B30.2 domain [19]. The distinctive structure of the PYD domain (amino acids 1–300), identified for the first time when the MEFV gene was cloned, was not analogous to any other protein domain known at the time; hence, it was named the PYD or PYRIN domain. Since its discovery, it has been found in more than 20 proteins regulating inflammation [20]. It is responsible for the homotypic interaction with ASC, an apoptosis-associated speck-like protein that promotes the activation of caspase-1 [21]. Typically, ASC oligomerises with one of the NLRP proteins and with procaspase-1 through homotypic CARD (Caspase recruitment domain) interactions to make up the inflammasome. This complex brings two molecules of precursor pro-caspase-1 into close proximity, leading to autocatalysis and, therefore, the release of the active catalytic p20 and p10 domains of caspase-1 [22]. Caspase-1, in turn, cleaves the pro-form of IL-1β into its active form (Figure 2). 

In order to better characterise FMF, researchers started to investigate the biological function of unmutated pyrin. In the past, depending on the experimental setting, pyrin was shown to both activate and inhibit the caspase-1/IL-1β signalling pathway [23,24,25]. The process of pyrin inflammasome inhibition has been described to depend on RhoA phosphorylation [19]. RhoA activates the serine–threonine kinases PKN1 and PKN2 that phosphorylate pyrin. Phosphorylated pyrin binds to regulatory proteins, such as 14-3-3, that avoid the formation of the pyrin inflammasome (Figure 2A). The toxins produced by bacteria are able to inactivate RhoA and, in turn, inhibit PKN1 and PKN2 activation, with the consequent dephosphorylation of pyrin (Figure 2B). Pyrin is, therefore, able to interact with ASC and caspase-1, forming the pyrin inflammasome, with the consequent cleavage and secretion of IL-1β (Figure 2B). FMF-associated mutations of the B30.2 domain make the protein less prone to phosphorylation, thus leading to constitutive activation of the pyrin inflammasome, influencing the interaction between the regulatory protein 14-3-3 and pyrin [19] (Figure 2B).

Due to the autosomal recessive mode of transmission, FMF was believed to be caused by a loss-of-function mutation of pyrin. However, pyrin knockout mice develop normally and do not exhibit an inflammatory phenotype. A further challenge to the loss-of-function theory is given by the fact that some individuals display the disease despite being heterozygous for one single mutation [26,27]. Moreover, asymptomatic carriers of MEFV mutations can have elevated acute-phase reactants [28]. Homozygous pyrin “knockin” mice harbouring mutant human B30.2 domains, but not pyrin-deficient, exhibited spontaneous inflammation similar to but more severe than human FMF [29]. Caspase-1 was constitutively activated in knockin macrophages, and active IL-1β was secreted after LPS stimulation, as observed in FMF patients. The inflammatory phenotype of knockin mice was reversed after crossing with IL-1 receptor-deficient or adaptor molecule ASC-deficient mice but not with NLRP3-deficient mice. These pivotal studies provide the final evidence for an ASC-dependent, NLRP3-independent inflammasome in which gain-of-function pyrin mutations cause FMF [29].

The clinical consequence of the dosage effect of MEFV variants was described in children with periodic fevers, in which the prevalence of FMF-related clinical manifestations was significantly correlated to the number and pathogenicity of MEFV variants carried by the patients [30]. Similarly, the degree to which IL-1β is over-secreted from FMF monocytes after LPS stimulation was proportional to the number and pathogenicity of MEFV variants carried by the patients [31].

Dependence from RhoA makes the pyrin inflammasome distinct from other inflammasomes that are activated by pattern-recognition receptors: it does not directly interact with PAMPs and DAMPs. Instead, it indirectly senses and responds to pathogen virulence factors that modify RhoA, acting as a molecular “guard” that senses alterations in the homeostasis of a cell [32].

The crucial role of the pyrin inflammasome in the response to pathogens inducing toxin release (such as *Y. pestis*) led to the fascinating hypothesis of a possible selective advantage for individual carriers of MEFV causative during plague times [33] (Figure 2C). In fact, the *Yersinia pestis* virulence factor, called YopM, stimulates the PKN-1/2-mediated phosphorylation of pyrin and thereby the inhibition of pyrin inflammasome reducing IL-1β secretion in response to infection [34]. In turn, MEFV pathogenic variants attenuate the pyrin–YopM interaction, thus interfering with the YopM-induced interleukin-1β suppression (Figure 2C). Leukocytes from FMF patients release heightened IL-1β, specifically in response to *Y. pestis*, as compared to healthy controls. *Y. pestis*-infected knockin mice for pathogenic MEFV variants exhibit IL-1-dependent increased survival relative to wild-type knockin mice. Thus, MEFV pathogenic mutations confer heightened resistance to *Y. pestis* [34].

### IL-1β

Interleukin-1 (IL-1) is one of the most potent pro-inflammatory cytokines. Two distinct ligands (IL-1α and IL-1β) bind the IL-1 type 1 receptor (IL-1R1) inducing a pro-inflammatory cascade leading to the production of mediators, such as prostaglandins, cytokines, and chemokines [35]. The IL-1α precursor is constitutively present in most epithelial cells and is fully active. On the contrary, IL-1β is synthesised as an inactive precursor, only after activation of the cells, typically after the stimulation of toll-like receptors. The activation of IL-1β is contingent on proteolytic cleavage by caspase-1 [36]. IL-1 is able to induce the myeloid differentiation primary response gene 88(MyD88), and therefore the translocation of active NF-κB to the nucleus. NF-κB-dependent genes, such as NLRP3, pro-IL-1β, pro-IL-18, and IL-6, are the mediators in this process. The central role of IL-1 in the innate inflammatory processes, and, therefore, in autoinflammatory diseases, explains the clinical success of IL-1 blocking agents in treating such conditions [37,38].

## 4. Genetics and Genotype–Phenotype Correlations

The MEFV gene is made up of 10 exons and is localised on chromosome 16p13.3. To date, more than 300 variants have been identified and reported in the INFEVERS database (Infevers, Sarrauste de Menthiere et al., http://fmf.igh.cnrs.fr/ISSAID/infevers/index.php (accessed 30 April 2023) (Figure 3); however, the relative frequencies and pathogenicity are not known for most of them. The hot spots of the FMF-causing MEFV variants were found on exon 2, at position 148, and on exon 10, at positions 680 and 694 [39,40]. In Turkey, the most frequent mutations are M694V, E148Q, M680I, and V726A [41]. In the Israeli community, the common mutation for non-Ashkenazi Jews is M694V (76.8%) [42] and E148Q for Ashkenazi Jews [43], whilst V726A is the most encountered mutation among Arabs. These last three variants represent probably the most ancient MEFV mutations, and it is calculated that their appearance in the Middle East (Mesopotamia) could be dated to more than 2500 years ago [44]. The hypothesis is that migrations of a few families around the Mediterranean basin during the centuries led to a founder effect. Further evidence for this phenomenon comes from a study on an isolated Jewish community in Palma de Mallorca, the “Chuetas”, formed by 18 families in whom more than 60 FMF patients have been diagnosed, and their genotypes overlapped with those observed in North African Jews [45]. 

The modification to the methionine residue at position 694 was described as a high penetrance mutation on the discovery of the pyrin gene in 1997. In addition, despite the disease being typically considered recessive, some patients with the classical FMF phenotype were reported to have a seemingly dominant transmission [46]. 

Indeed, as far as phenotype–genotype correlations are concerned, several observations showed that a more severe phenotype, with high fever, splenomegaly, and musculoskeletal manifestations is usually associated with high penetrance mutations [47,48], such as M694V, which also seems to confer a less favourable response to colchicine [49]. On the other hand, the low penetrance variant E148Q has been suggested to have an aggravating effect: dominant transmission when allelic to M694I with a second wild-type allele and amyloidosis when allelic to V726A with a second mutated allele [50]. The mild phenotype or incomplete penetrance has also been described in patients with K695R or P369S [40]. 

In the past, twin studies proved full concordance between monozygotic twins and a 30% concordance rate between dizygotic twins, with some degree of clinical variability [51]. However, in addition to the MEFV gene, some other genetic loci may impact the pathogenesis of the disease as modifiers, such as MICA (major histocompatibility complex class I chain-related gene A) [52] and polymorphisms of the SAA1 (serum amyloid A1) gene [53]. 

## 5. Clinical Features

The clinical presentation of FMF can be variable, likely depending on its genetic heterogeneity and environmental factors. However, the clinical picture is usually very suggestive of the underlying disease. It is typically characterised by recurrent episodes of fever and systemic inflammation with (pleural and abdominal) serositis and arthritis. Starting in childhood, patients present short-lasting, self-resolving attacks of fever and abdominal, chest, or joint pain with systemic inflammation [54]. Periodicity is not strict, and episodes may occur from once a week up to once every three to four months or more in untreated patients. These events are usually very disabling and in clear contrast with complete well-being in attack-free periods [55]. 

Several triggers have been identified for the attacks, such as stressful events [56], cold exposure [57], and the menstrual cycle in pubertal and post-pubertal females [58]. A prodrome was found to be a common manifestation of FMF, experienced by about 50% of the patients. Most commonly, it entails a sensation of general malaise and discomfort, including neurological manifestations such as headache [59] or abdominal pain. 

In 90% of cases, the disease onset is in childhood, where 65% of patients are less than 10 years old [60]. A young age of onset (<2 years) is associated with a more severe course of illness, higher penetrance mutations [61], and a more pronounced delay in diagnosis [62]. Fewer studies have investigated adult-onset FMF, but it has been reported to have a mean age of clinical onset of 32.5 years, milder symptoms, and fewer or no disease complications [63]. For these reasons, adults suspected to have FMF may also benefit from different diagnostic tools compared to paediatric populations [64].

Disease flares in FMF are typically associated with an elevation in the acute phase response (C-reactive protein, CRP, and serum amyloid A, SAA). Sometimes, high CRP and SAA levels are also found during the attack-free periods, indicating a still active disease and a strong risk factor for the development of amyloidosis due to chronic inflammation [65]. Another recently described, yet not characterised, player in the development of proteinuria and amyloidosis in FMF is high serum levels of galectin-3 (gal-3). Gal-3 is a b-galactoside-binding lectin highly expressed in innate immune cells and involved in the pro-inflammatory and pro-fibrotic pathways [66]. Galectin-3 could be used in the future as a prognostic biomarker for the development of renal damage in FMF and other conditions [67]. 

### 5.1. Fever

A fever is present in 96% of inflammatory episodes, ranging from 38 °C to 40 °C [68]. It appears suddenly and lasts from 12 to 72 h. The typical cycle displays a spontaneous and fast rising in temperature followed by a plateau and rapid decrease [1]. In young children, a fever may represent the unique disease manifestation at onset, with the subsequent development of a typical clinical presentation (including serositis) over the next 2.9 ± 2.2 years [61]. 

### 5.2. Abdominal Manifestations

Abdominal pain is extremely frequent during fever episodes, being reported in 94% of patients [6], and is secondary to a sterile inflammation in the peritoneum. Usually, the pain is severe and induces the patient into an antalgic position and bed rest. Sometimes, it may mimic an acute abdomen with rebound tenderness, reduced peristalsis, and distension and rigidity of abdominal muscles. Radiographic features may reveal small air-fluid levels and can wrongly guide caregivers toward an explorative laparoscopy and possibly an unnecessary appendicectomy or cholecystectomy. Indeed, appendectomy in FMF patients was found to be much higher than the reported rate in the general population (40% vs. 12–25%), whilst the number of non-inflamed appendectomies was much higher (80% vs. 20%) [69]. Most of the time, peritonitis is completely resolved within 2–3 days without sequelae. Constipation is often observed during the episode, while diarrhoea occurs in 10–20% of episodes. In addition, the described vomiting rate is approximately 30% in children [30]. Possible long-term complications consequent to repeated bouts of peritoneal inflammation are abdominal adhesions, leading to sub-occlusion, and infertility, which were frequently observed in the pre-colchicine era [70]. 

A palpable enlarged spleen is found according to different series variably in 10–60% of patients [30,55]. Splenomegaly not related to amyloidosis can be detected using ultrasonography and is usually the direct result of bowel sterile inflammation, which also causes abdominal micro-lymphadenopathy [71].

### 5.3. Pleurisy and Pericarditis

Pleural serositis is also for chest pain, which manifests with dyspnoea and bilateral respiratory and pleural auscultatory friction sounds in the involved site of the pleura [72]. The frequency changes in diverse study groups and ranges between 20 and 60%. An additional X-ray finding is a transient small pleural effusion, which resolves within 48 h after the episode [60]. 

Pericarditis is rare but more frequent than in the general population (around 7 per 1000 cases vs. 0.5) [73] and may present with retrosternal pain and ST abnormalities on electrocardiogram. It usually occurs years after the diagnosis, even though uniquely, it could be its first sign [74]. Interestingly, idiopathic recurrent acute pericarditis (IRAP) can be seen in both autoinflammatory (such as FMF and TRAPS) and autoimmune conditions (systemic lupus erythematosus, rheumatoid arthritis, progressive systemic sclerosis, and others) suggesting a mixed pathogenesis involving both innate and adaptive immune systems [75].

### 5.4. Musculoskeletal Symptoms

Joint manifestations are observed in 50% of cases and manifest as transient arthralgia or mono/oligoarthritis. Recurrent monoarthritis is the most common presentation, and it usually affects the knee, hip, and/or ankle [76]. The arthritis is typically associated with a robust inflammatory reaction with redness and swelling in the involved joint. Arthrocentesis results in an aseptic exudate with a high number of inflammatory cells. Synovitis usually resolves after 24–48 h, with the same evolution of abdominal or chest attacks with no sequelae. However, progressively destructive arthritis has been described in the literature in some cases [77]. A less common manifestation of joint involvement in FMF is spondyloarthropathy, which is often HLA–B27 negative. 

Myalgia may be associated with FMF, ranging from spontaneous generalised self-resolving muscle ache, exertional leg pain and, less commonly, protracted febrile myalgia. Exertional leg pain has been characterised as less intense post-exercise acidification on the muscle of FMF patients as compared to controls [78]. Protracted febrile myalgia syndrome (PFMS) is a rare complication of FMF, which cannot be prevented with colchicine. PFMS is a long-lasting (4–6 weeks), intense, and disabling muscle pain, usually in the lower limbs, associated with high-grade fever and high inflammatory parameters, but normal muscle enzymes and non-specific inflammatory changes in the EMG [1]. A high signal intensity distributed around myofascicles in the inflamed muscles can be detected using MRI [79]. It requires high-dosage cortisone treatment or anti-Il1 agents [80]. 

### 5.5. Other Manifestations

As for other serosal membranes, inflammation of the tunica vaginalis producing orchitis may be another event occurring in FMF patients. The frequency of acute scrotum has been reported as up to 9% in some studies [81]. Scrotal attacks are unilateral, self-limited, and painful with red swelling of the testicle lasting 48–72 h. The presence of high fever helps in the differential diagnosis of testicular torsion. 

Albeit rare (about 13% of patients in the Yildirim G. et al. series [82]), erysipelas-like erythema (ELE) is the most typical cutaneous manifestation of FMF, seen most frequently associated with arthritis as a comorbidity. It appears as tender, indurated, inflamed, and erythematous plaques, usually located over crural areas, the ankle joint, and the dorsum of the foot. Foot erythema is usually associated with ankle arthritis. It may be triggered by physical effort and subside spontaneously within 48 to 72 h of rest. Recurrent oral ulceration was found to be relatively frequent (10%) in FMF [83]. Interestingly, heterozygous mutations in exon 2 on the phosphorylation site of pyrin (c.726C > G; p.Ser242Arg) have been associated with a unique phenotype distinct from the typical FMF, characterised by severe acne and pyoderma gangrenosum, in the so-called pyrin-associated autoinflammatory diseases with neutrophilic dermatosis (PAAND) [32].

## 6. Associated Diseases

Several inflammatory and autoimmune conditions have been associated with FMF and MEFV gene mutations, probably due to common dysregulations in the immune system but also to the high frequency of MEFV carriers in some populations. IgA vasculitis is the most frequent in FMF patients (with a prevalence of 2.7–7% vs. 0.003–0.026% in controls), followed by polyarteritis nodosa (PAN) with a prevalence of 0.9–1.4% (vs. 0.0005–0.0031% in the general population) [84]. The demyelinating central nervous system disease multiple sclerosis (MS) is enriched in FMF patients compared to Israeli and Turk populations [85,86]. In this regard, homozygosity for the M694V MEFV mutation may be a genetic modifier in MS, aggravating the phenotype of MS. Recently, the frequency of ankylosing spondylitis (AS), IgA vasculitis, juvenile idiopathic arthritis (JIA), PAN, and MS was found to be increased in a big cohort of patients with FMF when compared with those in the literature [87]. Finally, a strong association with hidradenitis was also shown [88]. 

## 7. Diagnostic and Classification Criteria

The first set of diagnostic criteria was created for adults by experts in Tel Hashomer Hospital [60] (Table 1), and thirty years later, they were refined by Livneh et al. [89] (Table 2). However, a diagnostic standard with high specificity and sensitivity was also necessary for the paediatric population, with the aim of an early diagnosis. Indeed, in 2009, a new set of paediatric criteria was developed by Yalçinkaya and Özen [90] (Table 3). These were validated with a cohort of Turkish children, reaching a sensitivity and specificity of 88.8% and 92.2%, respectively, and also encompassing some clinical aspects that are typical of children as opposed to adults (inability to express pain location, different range in the duration of attacks, diagnosis prior to appendicectomy, etc.). 

The paediatric criteria yielded a better sensitivity but a poorer specificity than the previous criteria when applied to an international cohort of children from either European or Eastern Mediterranean regions [91]. Conversely, the Tel Hashomer criteria displayed the best specificity but poor sensitivity. A higher specificity was meant to minimise the diagnostic failure or delay, although FMF diagnosis still needed to be refined with the inclusion of genetic data [92].

Indeed, in 2015, a group of experts built a set of evidence-based recommendations using a systematic literature review [93]. During the consensus meeting, the specialists confirmed the literature and concluded that FMF is a clinical diagnosis, which can be supported but not necessarily excluded with genetic testing (Strength B). This statement is currently a matter of debate since many authors believe that the term FMF should be applied only to patients carrying MEFV mutations (Ben-Chetrit), as FMF is a genetic condition.

In this line, the new evidence-based Eurofever/PRINTO classification criteria, developed for inherited recurrent fevers in 2019, includes for the first time an association between clinical and genetic variables, resulting in a high sensitivity and high specificity classification tool [94] (Table 4). These criteria can be used to differentiate among different inflammatory conditions and, despite being classification criteria, are mainly used for research purposes (clinical trials, translational studies). They may provide both clinical-based and genetic-based guides useful for the diagnostic orientation of FMF in clinical practice. 

## 8. Interpretation of MEFV Gene Variants

In routine practice, when the patient’s symptoms are consistent with the diagnosis of FMF, genetic tests are suggested. For the interpretation of variants resulting from molecular analysis, a committee has developed guidelines classifying genes as: (a) clearly or likely pathogenic; (b) variants of unknown significance or VOUS; or (c) clearly or likely benign [95]. 

M694V is considered a very severe mutation, and if present in homozygosity, even asymptomatic individuals should be considered for treatment (Strength A, [93]). M694V, V726A, M694I, M680I, and E148Q account for 70–80% of the cases in Mediterranean countries [96]. However, in cases where an uncommon variant is identified, physicians and molecular geneticists can utilise the INFEVER database. INFEVERS (Internet Fevers), created in 2002, is an online database that documents all the information available on mutations in autoinflammatory disorders-related genes (http://fmf.igh.cnrs.fr/ISSAID/infevers/index.php (accessed 30 April 2023)) [97]. It was conceived as a universal tool to gather and share all data on the mutational spectrum of HRF genes in a centralised location to highlight information that can be missed if reported separately. More specifically, it aims to overcome the challenges in interpreting VOUS with a low frequency which may function as susceptibility alleles to inflammation or new and rare genetic variants associated with a clear phenotype [98]. To date, this website documents more than 350 MEFV variants together with classification status (benign, likely benign, VOUS, likely pathogenic, pathogenic) and the centre that made the notification (Figure 3). The information provided in INFEVER can be complemented with other databases on the clinical significance of human genetic variants (ClinVar), and some in silico prediction tools (AGVGD, Sorts Intolerant from Tolerant, Polyphen-2, and Combined Annotation-Dependent Depletion score).

## 9. Treatment

### 9.1. Colchicine: Mechanism of Action

Colchicine is the oldest known drug still used today [99]. It is an alkaloid derivative of the plant *Colchicum autumnale*.

In the 1960s, colchicine’s ability to bind microtubules was discovered, revealing its antimitotic action [100]. However, colchicine’s action in FMF still remains to be elucidated to some degree. Microtubules, the molecular targets of colchicine, operate pleiotropically within the cell, governing intracellular organelle and vesicle transport, secretion of cytokines and chemokines, cellular migration and division, and regulating gene expression [101]. To do so, microtubules act in a dynamic fashion, changing their shape, by adding and losing tubulin heterodimers, in a continuous equilibrium between extension and shrinkage. Colchicine blocks polymer elongation, effectively inhibiting microtubule properties [102]. Colchicine action leads to impaired neutrophil chemotaxis, by diminishing the expression of L-selectin on neutrophils cell membranes and E-selectin on endothelial cells [103], and neutrophil function, by inhibiting superoxide production [104]. Moreover, colchicine dampens the activation of macrophages and the degranulation of mast cells [105] and interferes with TNF-α pro-inflammatory actions [106] and, thereby, with the NF-kB signalling cascade [107].

In addition to the indirect action on chemotaxis, motility, and stimulation of leukocytes, colchicine has been demonstrated to inhibit the NLRP3 inflammasome, thereby suppressing caspase-1 activation in gout [108]. Additionally, it may also have a distinct inhibitory function on the pyrin inflammasome, explaining its specific effect on FMF and not in other autoinflammatory diseases. In fact, by acting on microtubes, colchicine is thought to activate—or release from inhibition—RhoA, resulting in suppression by phosphorylation of the pyrin inflammasome assembly [19]. 

### 9.2. Colchicine: Metabolism and Toxicity

Colchicine is absorbed in the jejunum and ileum. Its bioavailability depends on the hepatic, biliary, and luminal intestinal multidrug transporter P-glycoprotein 1 (PGY 1). The altered expression of this transporter protein may signify suboptimal therapeutic effects or drug toxicity [109]. Colchicine is eliminated with biliary excretion and through the stool [110]. A significant role in colchicine metabolism is played by enteric and hepatic cytochrome P450 3A4 (CYP450 3A4), which catalyses the demethylation of colchicine into the inactive metabolites 2- and 3-demethtylcolchicine. This is relevant because drugs modifying the activity of this cytochrome can result in colchicine-induced toxicity. Finally, 20% of drug disposal is accounted for by kidney secretion.

Colchicine is a safe drug that has been used for a long time; however, it has a narrow therapeutic index and its commonest side effects may occur even at treatment dosages. These are mainly gastrointestinal: cramping, abdominal pain, hyperperistalsis, diarrhoea, and vomiting. These manifestations appear in 10–15% of patients and tend to resolve after a period of treatment or dose reduction [111]. Blood dyscrasias and neuropathies are features of a chronic-type overdose [112]. 

High colchicine concentrations are extremely toxic, leading to severe microtubule disarrangement. The affected cells experience a halt in protein assembly, endocytosis, exocytosis, cellular motility, mitosis, cardiac myocyte conduction, and contractility [113]. The accumulation of these mechanisms may lead to multi-organ dysfunction and failure, consisting of three overlapping phases. Around 10–24 h from ingestion, severe gastrointestinal manifestations appear. Then, 24 h to 7 days later, multi-organ dysfunction takes place: bone marrow failure, renal insufficiency, adult respiratory distress syndrome (ARDS), arrhythmias, disseminated intravascular coagulation (DIC), neuromuscular disturbances, and alopecia are seen [114]. If the patient survives, recovery occurs in a week or so [111]. 

Colchicine overdose can occur when daily doses are not adjusted for reduced renal function or interacting medications. Indeed, simultaneous use of CYP3A4 inhibitors/competitors, including clarithromycin and erythromycin, many HIV medications, calcium channel blockers and azole antifungals, or P-gp inhibitors/competitors such as ciclosporin and ranolazine, can increase colchicine concentration [109]. 

Colchicine usage has also been associated with an increase in liver function enzymes, for reasons that are not always clear. 

Finally, in addition to the abdominal side effects of colchicine, evidence points toward the exitance of drug-induced lactose intolerance in treated FMF patients, which can be corrected with a lactose-free diet [115]. 

The risk of colchicine-driven oligo-/azoospermia is still a matter of debate. Probably, the frequency of azoospermia is influenced more by the underlying pathology [116], or by the presence of testicular amyloidosis [117], rather than the drug itself. Indeed, healthy volunteers do not experience infertility under colchicine treatment [118], and colchicine does not cause reduced sperm motility [119]. 

Concerning female fertility, colchicine therapy throughout pregnancy seems to carry no substantial teratogenic or mutagenic risk when used at recommended doses [120]. Additionally, colchicine was shown not to be associated with a higher rate of miscarriage, stillbirth, low birth weight, or early delivery [121]. 

### 9.3. Colchicine: Management in Familial Mediterranean Fever 

Early independent RCTs demonstrated that daily colchicine is highly effective for preventing attacks in this disorder in a dose-related fashion [4,5,122]. 

According to the ongoing EULAR recommendations [123], the “starting dose” of colchicine was defined as ≤0.5 mg per day for children less than 5 years old, 0.5 mg per day for 5–10-year-old children, and 1 mg for those aged 10–18 years and for adults. 

The dose should be guided mainly by the occurrence of clinical symptoms and serological inflammation, with the indication to increase the dose of 0.25 mg/day in a stepwise fashion until the maximally tolerated dose [123]. The maximal dose is considered to be 1 mg/day for children aged less than 5 years, 2 mg/day for pre-pubertal children, and 3 mg/day for post-pubertal children and adults [124].

Very few studies on colchicine dose per kilo have been completed. Using a cohort of children, the mean effective colchicine dosage was calculated to be 1.46 ± 0.41 mg/$m^{2}$/day or 0.05 ± 0.02 mg/kg/day for children <5 years; 1.19 ± 0.3 mg/$m^{2}$/day or 0.03 ± 0.01 mg/kg/day for children 6–11 years old; and 0.84 ± 0.2 mg/$m^{2}$/day or 0.027 ± 0.01 mg/kg/day for children 11–15 years old [125] (mean dose of the whole group was 1.16 ± 0.45 mg/$m^{2}$/day or 0.03 ± 0.02 mg/kg/day). These findings are consistent with a later study evaluating the influence of anthropometric parameters on colchicine dosage: young children received higher doses of colchicine according to their body weight as compared with older children. Furthermore, this analysis revealed that colchicine intake was best correlated with body surface area (~1.03 mg/$m^{2}$/day) [126].

The optimal treatment dose still remains to be defined; however, in any case, colchicine doses should not—and usually does not—reach values of 0.5–0.8 mg/kg, which are highly toxic or lethal (>0.8 mg/kg) [111].

## 10. Colchicine Resistance

Despite optimal treatment, around 5% of patients do not respond at all to the maximally tolerated dose of colchicine. A higher percentage (from 20 to 40%) of patients display an incomplete response, with a reduction but not complete control of fever episodes.

In 2016, the European League Against Rheumatism (EULAR), in its recommendations for the management of FMF, defined resistance as one or more attacks per month in compliant patients who had been receiving the maximally tolerated dose for at least 6 months [123]. More recently, a consensus of experts updated these recommendations with several statements, including some recommendations on adherence, dose adjustment criteria, and quality of life [127]. The conclusions of the consensus were schematically reported in seven statements (Table 5).

A recent multicentre and longitudinal study provided the possibility to verify in real life the actual impact of these statements concerning colchicine management (Table 5). In this study, 221 (125 children, 96 adults) Italian FMF patients treated with colchicine were followed for a median follow-up of 3.7 years [128]. Compliance to the drug was generally high (Figure 4). A complete response (absence of any fever episodes and persistent normalisation of inflammatory parameters) was achieved in 55.2% of patients. As expected, 7.7% of patients were classified as resistant (≥one episode/month) according to the EULAR recommendations [128]. However, almost 30% of the patients were classified as partial responders since they presented a significant reduction in the number of fever episodes/year with fewer than one episode per month. Out of the partial responders, around 70% of them displayed a few episodes per year (from one to four); however, a relevant percentage (≈30%) displayed a rather high number of episodes per year (from five to eight) (Figure 5). Interestingly, in all age groups, a relevant proportion (almost 20%) of patients with residual disease activity were still on their colchicine starting dose (Figure 6). The maximal recommended colchicine dose (1 to 3 mg/day according to the age group) was not achieved in any of the resistant patients or patients with an incomplete response. This study provides evidence for the general treatment with colchicine in real life [128]. 

On the other hand, almost 30% of patients with a partial response, who were not considered resistant according to the current EULAR recommendations, reported a limitation in at least one item related to their quality of life, with a limitation in daily activities/presence at school or work or the presence of chronic pain or fatigue [128].

## 11. Interlukin-1 Inhibitors

Given that pyrin is implicated in the synthesis of IL-1, which is probably the strongest inducer of inflammation, its inhibition represents a new approach to treat FMF.

Three different types of anti-IL-1 treatments are available. Anakinra is a human recombinant non-glycosylated analogue of the IL-1 receptor antagonist (IL-1Ra) [129]. Rilonacept is a fusion protein engineered to contain the extracellular domain of type I IL-1 receptor fused with the Fc portion of IgG1. Canakinumab is a fully humanised monoclonal antibody of the class IgG1 that acts specifically against IL-1 beta [130]. 

### 11.1. Anakinra

Being an analogue of the receptor IL-1Ra, anakinra can competitively inhibit the binding of both IL-1α and β; however, there is no significant difference in the biological activities of either cytokine. It is administered as a daily subcutaneous injection [131]. 

Over the last several years, evidence for the important role of anakinra in the prevention of serositis attacks in patients with colchicine-resistant FMF has emerged. Anakinra was the drug that showed a higher degree of efficacy in colchicine-resistant patients in one of the first reports from the Eurofever registry [132]. The first RCT on the efficacy of anti-IL1 treatments was conducted in 2017, showing the efficacy and safety of anakinra for the treatment of colchicine-resistant FMF compared to a placebo [133]. The mean and SD were 1.7 ± 1.7 attacks per patient per month in the anakinra group versus 3.5 ± 1.9 attacks in the placebo group (P ± 0.037). However, considering site-specific attacks, the difference between the anakinra and placebo groups reached significance only for attacks in the joints. In this respect, anakinra may be complementary to colchicine, which often fails to prevent attacks in the joints while suppressing activity in other sites. There were no severe adverse events over a 20-month follow-up period. 

On the other hand, several case studies reported an improvement in renal function for patients with amyloidosis following anakinra treatment [129,134]. The use of anakinra during pregnancy was shown to be safe [135,136] and is currently recommended in colchicine-resistant women. 

The most common side effect is injection site reaction [137]. Albeit uncomfortable, these usually resolve within 2–3 weeks of treatment initiation; however, they may be so severe to prompt a patient to interrupt treatment [138]. 

As for other anti-cytokine treatments, a major concern is the risk of infection. Nevertheless, in comparison to other biologic agents, anakinra has an unparalleled safety benefit deriving from a short half-life, and the effect duration and has demonstrated a remarkable record of safety [139].

### 11.2. Rilonacept

Rilonacept is a very high affinity “cytokine trap” consisting of fusions between the constant region of IgG and the extracellular domains of two distinct cytokine receptor components involved in binding the cytokine [140]. It is administered weekly with an injection.

The first randomised placebo-controlled study on FMF with an anti-IL-1 agent was performed with rilonacept. The study included 12 patients and rilonacept significantly reduced the number of FMF attacks and had an acceptable safety profile, with no serious side effects associated with this drug [141]. 

### 11.3. Canakinumab

Canakinumab is the only FDA-approved cytokine blocker for the treatment of colchicine-resistant FMF in the United States [142]. Its long half-life allows a monthly subcutaneous administration.

The first report in the literature on the successful administration of canakinumab in a patient with FMF and chronic arthritis after failing anakinra, etanercept and low-dose prednisone, and methotrexate was published in 2011 [143]. A significant decrease in proteinuria in the amyloidosis-complicated FMF patients was observed [144]. All the series reported that patients benefit from canakinumab [145,146,147] and others, also in terms of quality of life [63].

The efficacy of the treatment was confirmed when randomised against a placebo in a cohort of colchicine-resistant FMF patients together with TRAPS and mevalonate kinase deficiency (MKD) patients [148]. A complete response occurred in 71% of FMF patients when treated with canakinumab (150 or 300 mg subcutaneously every 4 weeks). Patients who did not have a complete response had a lower number of days with fever per year. When an extended dosing regimen (canakinumab every 8 weeks) was evaluated, the absence of flares was maintained in approximately half the patients with colchicine-resistant FMF. In this study, no deaths, opportunistic infections, or cancers were reported.

In all three cohorts, infections were more numerous in the canakinumab group than in the placebo group, serious infections being rare (7.4 per 100 patient-years). Three patients had to discontinue treatment because of neutropenia [148]. The long-term efficacy and safety of canakinumab in the phase 3 cluster trial of the same study were recently reported [149]. 

## 12. Anti-IL-6 Drugs

IL-6 is elevated in the serum of FMF patients during attacks, and its potential as a biomarker to distinguish between acute phase and remission [150] and drug target was investigated. Tocilizumab (TCZ) is a humanised monoclonal anti-IL-6 receptor antibody, binding to soluble and membrane receptors and down-regulating IL-6 synthesis, and as a consequence, possibly suppressing SAA production. Indeed, the result from a series of 12 patients with AA amyloidosis secondary to FMF treated with TCZ, showed an improvement in attacks [151].

The long-term safety of TCZ is now being investigated in a Japanese multicentre placebo-controlled, randomised, double-blind trial on colchicine-resistant and colchicine-intolerant FMF [152].

### Conclusive Remarks and Future Perspective

In conclusion, Familial Mediterranean Fever (FMF) is the first inflammatory condition for which a causative gene was identified and represents a prototype of a monogenic autoinflammatory disease condition. In recent years, significant progress has been made in understanding the pathogenic mechanisms related to this condition. Early diagnosis and prompt treatment with colchicine can effectively manage symptoms and prevent complications.

Future research efforts should focus on developing more effective therapies for FMF patients who are unresponsive to colchicine. Further studies are also needed to identify new genetic mutations that contribute to FMF and to explore the possible association between FMF and other diseases. Moreover, the development of biomarkers for monitoring disease progression and response to therapy would be beneficial for improving the clinical management of FMF. In addition, genetic counselling and family screening programs should be implemented to identify asymptomatic carriers and prevent the transmission of the disease to future generations. In summary, while significant progress has been made in understanding and treating FMF, there is still much to be done to improve patient outcomes and quality of life. With continued research efforts and collaboration among healthcare professionals, we can work toward better management and ultimately a cure for this condition.

## Figures and Tables

**Figure 1 ijms-24-09584-f001:**
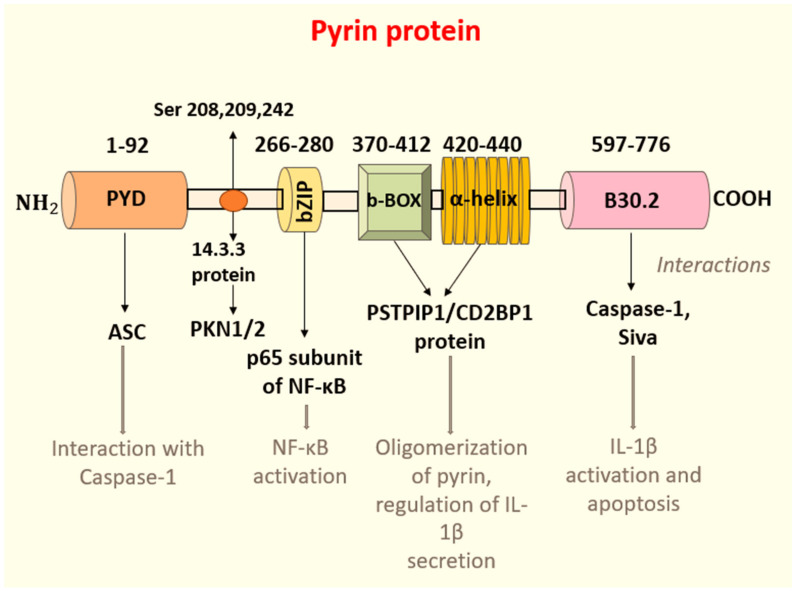
Schematic representation showing the pyrin protein. Pyrin is an approximately 95 kDa protein made up of five domains: a PYD or PYRIN domain (1–92), bZIP transcription factor domain (266–280), B–box zinc finger (370–412), α–helix coiled–coil domain (420–440), and a C–terminal B30.2 domain (597–776). The N–terminal PYD domain is responsible for the interaction with ASC (apoptosis–associated speck–like protein with a caspase recruitment domain), which, in turn, mediates the CARD (caspase recruitment domain) CARD homotypic interface with caspase–1. The bZIP transcription factor basic domain promotes NF–kB activation via the interaction with its subunit p65. The B–box zinc finger and the α–helix domain are involved in the oligomerization of pyrin and the regulation of IL–1β secretion. The B30.2 domain harbours most of the FMF–causing mutation and is functionally important in the activation of the pyrin inflammasome. B30.2 interacts with caspase–1 and pro–apoptotic protein Siva. Three residues of pyrin, serines 208, 209, and 242, are responsible for interacting with the 14.3.3 regulatory molecule that participates in the phosphorylation via PKN 1/2 (serine/threonine protein kinase C–related kinase 1/2).

**Figure 2 ijms-24-09584-f002:**
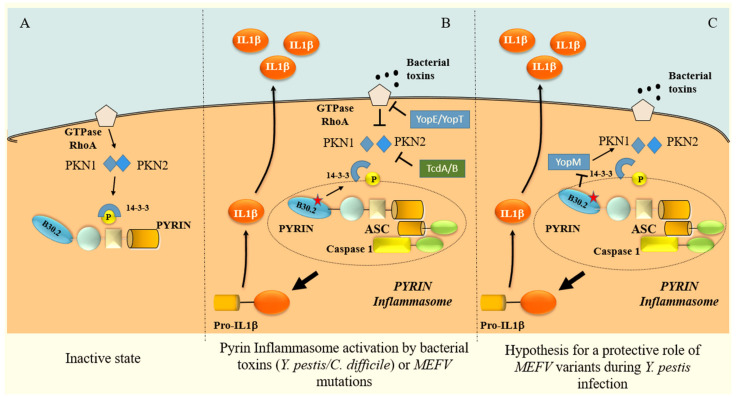
(**A**) Phosphorylated pyrin in an inactive state. In the steady-state condition, RhoA promotes the inactive configuration of pyrin by inducing its phosphorylation, mediated by the serine–threonine kinases PKN1 and PKN2. B30.2 domain mutations are likely to control pyrin phosphorylation by inhibiting the binding of kinases to pyrin [19]. (**B**) Pyrin inflammasome assembly promoted by exotoxins or pathogenic MEVF mutations. Toxins produced by some bacteria (i.e., YopE and YopT from *Yersinia pestis*) directly inactivate RhoA; other toxins, such as TcdA/B from C. difficile, directly inactivate PKN1 and PKN2. The final result is the inhibition of PKN1 and PKN2 activation with the consequent dephosphorylation of pyrin. Pathogenic MEFV mutations of exon 10 (B30.2 domain) interfere with the binding of the 14-3-3 protein to pyrin, leading the pyrin protein to be more susceptible to dephosphorylation. Dephosphorylated pyrin is active and able to interact with ASC and caspase-1, forming the pyrin inflammasome. IL-1β is cleaved to its active form as a result of the autocatalysis and activation of two precursors molecules of caspase-1. (**C**) Hypothesis for the possible protective role of the MEFV mutation during *Yersinia pestis* infection. The Y. pestis-induced virulence factor YopM directly inhibits pyrin inflammasome formation by promoting PNK1/2-mediated pyrin phosphorylation. MEFV mutations on the B30.2 domain attenuate the pyrin–YopM interaction, thus interfering with YopM anti-inflammatory activity.

**Figure 3 ijms-24-09584-f003:**
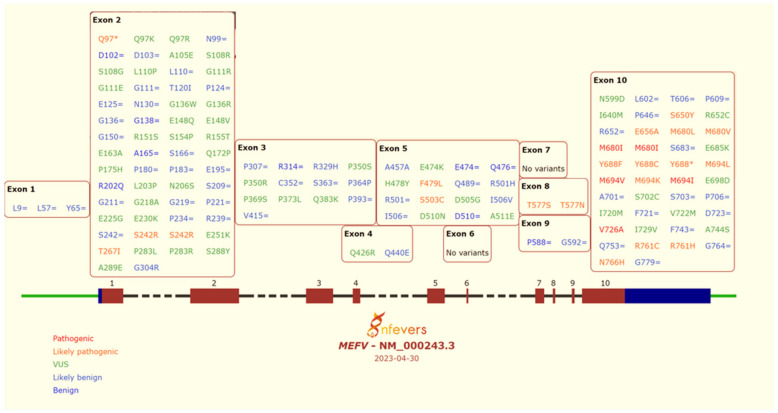
MEFV mutational spectrum based on the free source INFEVER online database. In red and orange are marked, respectively, pathogenic and likely pathogenic mutations; in green are marked VUS (variance of unknown significance); and in light blue and blue are marked likely benign and benign variants. Hot spots for pathogenic mutations are found on exons 2 and 10.

**Figure 4 ijms-24-09584-f004:**
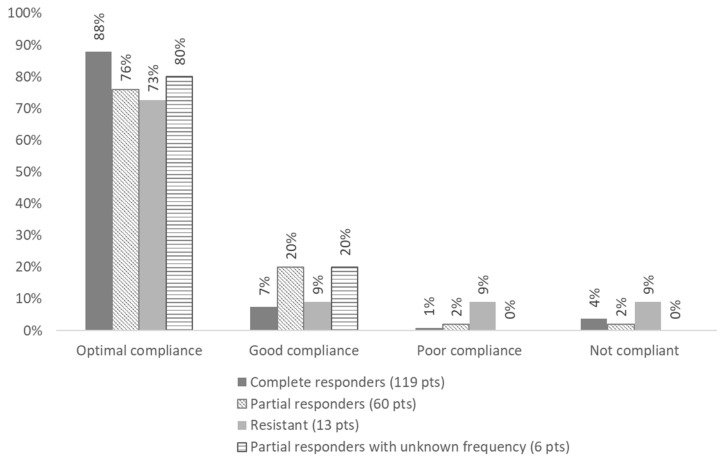
Compliance with colchicine treatment according to disease activity. Optimal compliance (compliant with >90% of prescriptions). Good compliance (compliant with 50–89% of prescriptions). Poor compliance (compliant with less than 50% of prescriptions) [125].

**Figure 5 ijms-24-09584-f005:**
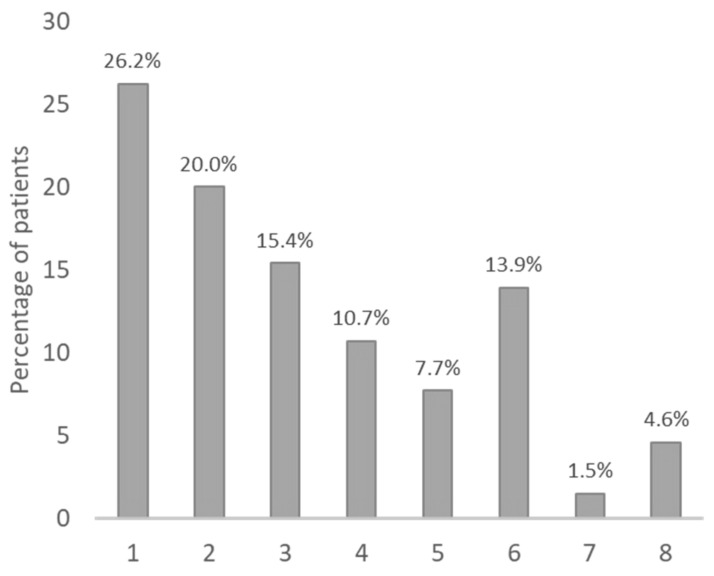
Number of episodes/year in patients with incomplete response to colchicine and less than one episode/month (partial responders) [128].

**Figure 6 ijms-24-09584-f006:**
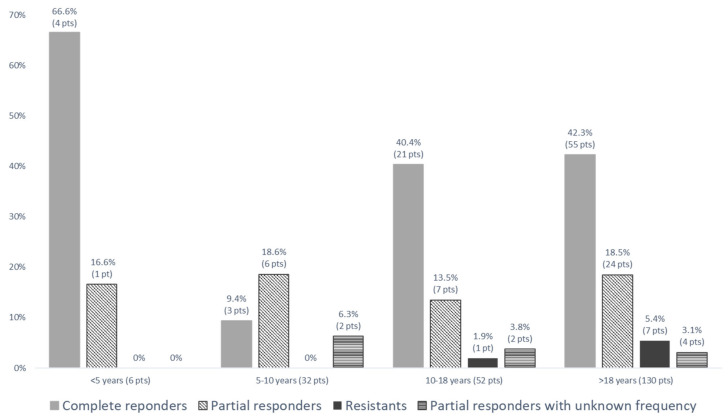
Colchicine response by age group in patients still receiving equal or less than the starting dose [128].

**Table 1 ijms-24-09584-t001:** Tel Hashomer criteria for FMF diagnosis (adult criteria).

Tel Hashomer criteria (at least two major criteria or one major criterion plus two minor criteria)
Major criteria
- Recurrent febrile episodes accompanied by peritonitis, synovitis, and pleurisy. - AA amyloidosis without predisposing disease. - Favourable response to continuous colchicine administration.
Minor criteria
- Recurrent febrile episodes. - Erysipelas-like erythema. - FMF diagnosed in a first-degree relative.

**Table 2 ijms-24-09584-t002:** Livneh criteria for FMF diagnosis.

Livneh criteria (at least one major criterion, or two minor criteria, or one minor criterion plus at least five supportive criteria, or one minor criterion plus at least four of the “first” five supportive criteria)
Major criteria
- Typical attack * of generalised peritonitis. - Typical attack * of unilateral pleuritis/pericarditis. - Typical attack * of monoarthritis. - Presence of fever alone (rectal temperature of 38 °C or higher).
Minor criteria
- Incomplete attack ** involving the abdomen. - Incomplete attack ** involving the chest. - Incomplete attack ** involving one large joint. - Exertional leg pain. - Favourable response to colchicine.
Supportive criteria
- Family history of FMF. - Appropriate ethnic origin. - Age less than 20 years at disease onset. - Severity of attacks requiring bed rest. - Spontaneous remission of symptoms. - Presence of symptom-free intervals. - Transient elevation of inflammatory markers. - Episodic proteinuria or hematuria. - Non-productive laparotomy with removal of a “white” appendix. - Consanguinity of parents.

* Typical attacks are defined as recurrent (≥ three of the same type), febrile (rectal temperature of 38 °C or higher), and short (lasting between 12 h and 3 days).** Incomplete attacks are defined as painful and recurrent flares that differ from typical attacks in one or two features as follows: (1) normal temperature or lower than 38 °C, (2) attacks longer than 1 week or shorter than 6 h, (3) no signs of peritonitis recorded during an acute abdominal complaint, (4) the abdominal attacks are localised, and (5) arthritis involves joints other than those specified [86].

**Table 3 ijms-24-09584-t003:** Yalcinkaya–Ozen criteria for FMF diagnosis (childhood criteria).

Yalcinkaya-Ozen (childhood) criteria (at least two out of five criteria)
- Fever (axillary temperature > 38 °C, 6–72 h of duration, ≥ three attacks). - Abdominal pain (6–72 h duration, ≥ three attacks). - Chest pain (6–72 h duration, ≥ three attacks). - Oligoarthritis (6–72 h duration, ≥ three attacks). - Family history of FMF.

**Table 4 ijms-24-09584-t004:** Eurofever/PRINTO classification criteria for FMF.

Eurofever/PRINTO Classification Criteria for FMF
Presence of confirmatory MEFV genotype * and at least one, or not confirmatory MEFV genotype ** and at least two, of the following:
- Duration of episodes 1–3 days. - Arthritis. - Chest pain. - Abdominal pain.

A patient with (1) evidence suggesting elevation of acute phase reactants (ESR or CRP or SAA) in correspondence to clinical flares and (2) careful consideration of possible confounding diseases (neoplasms, infections, autoimmune conditions, other inborn errors of immunity) and a reasonable period of recurrent disease activity (at least 6 months) is classified as having hereditary recurrent fever if the criteria are met. * Pathogenic or likely pathogenic variants (heterozygous in AD diseases, homozygous or in trans (or biallelic) compound heterozygous in AR diseases). ** In the trans compound, heterozygous for one pathogenic MEFV variant and one VUS, or biallelic VUS, or heterozygous for one pathogenic MEFV variant [91].

**Table 5 ijms-24-09584-t005:** The 2021 Delphi consensus final statements by Özen et al. on colchicine resistance/intolerance and their application in a national multicentre longitudinal study.

	Delphi Expert Consensus Statements (Özen et al., 2021 [124])	Results of a National Multicentre Longitudinal Study (Bustaffa et al., 2021 [125])
**Adherence**	Statement 1: Colchicine is the drug of choice for the treatment of FMF, and adherence is a critical issue. For the following statements, it is assumed that the patient is adherent to their prescribed colchicine treatment.	Overall, 83.8% displayed optimal adherence (>90% of prescription); 10.6% displayed good adherence (50–89% of prescriptions); 2.0% displayed poor adherence (<50% of prescriptions); and 3.5% displayed no adherence.
**Dose adjustment criteria**	Statement 2: When utilising colchicine to treat FMF, it is recommended to adjust the dose based on disease activity, with the adjustment of maximal dose for children depending on age (and weight).	The percentage of patients taking a colchicine dose without adjustments were: - Patients < 5 years: 71.3%; - Patients 5–10 years: 35.3%; - Patients aged 10–18 years: 57.0%; - Adults: 67.1%.
**Recommended maximum colchicine dose**	Statement 3: The maximum recommended colchicine dose for the treatment of FMF is 1–3 mg per day, depending on age and weight, limited by signs of toxicity and tolerability (see below).	No patient reached the maximum recommended dose.
**Resistance to colchicine**	Statement 4: For a patient receiving the maximum tolerated dose of colchicine, resistance to colchicine is defined as ongoing disease activity (as reflected by either recurrent clinical attacks (average one or more attacks per month over a 3-month period), or persistently elevated CRP or SAA in between attacks (depending on which is available locally)) in the absence of any other plausible explanation.	Resistance was be defined as the persistence of fever attacks, despite optimal treatment. Overall, 54.2% patients had a complete disease control; 30.1% patients had < one episode/month for 3 months; 8.5% had ≥ one episode/month for 3 month; and 7.2% had persisting disease with an unknown frequency of attacks.
**Inclusion of secondary amyloidosis in the definition of colchicine resistance**	Statement 5: AA amyloidosis develops as a consequence of persistent inflammation, which may be a complication of colchicine resistance.	Five adult patients (2.1%) displayed amyloidosis, two of which were prescribed anti-Il1 treatment.
**Colchicine intolerance**	Statement 6: Colchicine intolerance, which generally manifests as mild gastrointestinal symptoms (such as diarrhoea and nausea), is common but can limit the ability to achieve or maintain the effective dose. Dose-limiting toxicity is rare and may include serious gastrointestinal manifestations, such as persistent diarrhoea, elevated liver enzymes, leukopenia, azoospermia, neuromyopathy, etc.	Eight patients (3.4%) with follow-up had persistent manifestations of intolerance to colchicine. No patient experienced real toxicity.
**Patient quality of life and self-reported outcome**	Statement 7: Active disease and intolerance to colchicine affect quality of life.	Overall, 20.1% of patients experienced fatigue or chronic pain, 26.6% experienced limitations in daily activities, and 19.6% lost school/work days.

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
