# Peer review of "An Update on Familial Mediterranean Fever"

_ijms, 2023, doi:10.3390/ijms24119584_

Round 1

Reviewer 1 Report

Dear Dr. Lancieri, 

Thank you for submitting your manuscript, which was very interesting and well-written. I think it will be a valuable contribution to the literature with some work. I am sending my suggestions in the attached PDF file, which includes 128 comments of mine. Most of the comments are related to typos or grammar mistakes. There are three comments relating to the content. I hope you find my suggestions helpful. 

Sincerely, 

Author Response

Dear Editor

Please find enclosed the revised version of our manuscript “An update on Familial Mediterranean Fever”

We thanks the reviewers for the useful comments and suggestions. Below find your reply to their comments

Reviewer 1

Thank you for submitting your manuscript, which was very interesting and well-written. I think it will be a valuable contribution to the literature with some work. I am sending my suggestions in the attached PDF file, which includes 128 comments of mine. Most of the comments are related to typos or grammar mistakes. There are three comments relating to the content. I hope you find my suggestions helpful.

Authors’ reply: Thank you very much for your comments, we found them very valuable. We provided to correct all the typos and grammar mistakes that you have highlighted and rephrase the sentences where you suggested it was not clear the meaning. Thank you

Reviewer 2 Report

The study titled “an update on familial Mediterranean fever” is interesting and well-written. However, I am not entirely sure if the article fits in the scope of the journal. The manuscript mostly discusses details on the fever, clinical features and therapeutics. The manuscript deals with the disease and its clinical features but a molecular analysis of the disease is direly missing. I have provided a few suggestions and comments that the authors can consider improving the quality of the manuscript.

1.      Section 1: “ in 1997 by and International….” The meaning is incorrect.

2.      “ In 1945 Siegal, …..” the sentence needs to be rephrased for proper meaning.

3.      Page 2: : how children leaving in…” Did the authors mean “living”?

4.      The authors should give a detailed description of the mutations observed in the MEFV gene. In addition, there should be a sequence-structure relationship analysis of the mutations. More importantly, the structure of pyrin molecule should be provided along with the mutated structures.

5.      Figure 1 caption is not aligned. In the figure, beta is given as “b” and this needs to be changed to the proper symbol. The figure caption for A and B is missing. Different shapes aren’t properly annotated, eg. Square and circle in pyrin molecules.

6.      Page 2: “ It contains N-terminal PYD domain…” What is the pyd domain? The authors need to explain the pyd domain. The authors need to provide a domain picture of the pyrin protein with their respective functions.

7.      Page 4: “Pyrin interacts with the Yersinia pestis……” The authors need to provide an image of the mechanism explained in the paragraph.  

8.      It is not clear whether the MEFV gene is dominant or recessive. The authors need to be clear on this part citing proper references.

9.      The mechanism of CRP and SAA levels during the disease needs to be explained in detail with a figure.

10.   The mechanism of gal-2 and how the same can be a biomarker needs to be explained in detail with a figure.

11.   Beta should be given as a symbol in all instances.

12.   Page 7: x-ray should be given as X-ray.

13.   Page 9: position 242 should be given as single letter residue and the residue number.

14.   Page 12: Strength B: What is this?

15.   Page 13: Strength A: What is this?

16.   Page 17, 18 and 19: The figure numbers are missing and are given as letters. Why?

17.   Page 13: “Colchicum Autumnale”…a should be in lower caps.

18.   Section 9.2: The section is unnecessary and does not add significantly to the review. This portion should be excluded.

19.   Section 9.3: “According to the ongoing….” What do the authors mean by 0.5mg/die? What does die mean in the unit?

20.   Figures Z, Y and XX are based on the published data and so proper citation must be provided in the figure captions.

21.   A future perspective of the review should be given along with concluding remarks.

22.   The binding and mechanism of the IL-1 antibodies should be given with proper structures and interaction data.

23.   There are several grammatical and typographical errors that the authors need to proofread. The representation of scientific data should be uniform and according to the norms.

Author Response

Reviewer 2

The study titled “an update on familial Mediterranean fever” is interesting and well-written. However, I am not entirely sure if the article fits in the scope of the journal. The manuscript mostly discusses details on the fever, clinical features and therapeutics. The manuscript deals with the disease and its clinical features but a molecular analysis of the disease is direly missing. I have provided a few suggestions and comments that the authors can consider improving the quality of the manuscript.

  1. Section 1: “ in 1997 by and International….” The meaning is incorrect.

Authors’ reply Thank you for the suggestion, we have corrected it.

  1. “ In 1945 Siegal, …..” the sentence needs to be rephrased for proper meaning.

      Authors’ reply:  Thank you for the suggestion, we rephrased it as In 1945, Siegal defined “benign paroxysmal peritonitis” as an under-diagnosed and “unusual clinical syndrome” in himself and other patients…

  1. Page 2: : how children leaving in…” Did the authors mean “living”?

Authors’ reply : Yes, we corrected with “living” in the text.

  1. The authors should give a detailed description of the mutations observed in the MEFV gene. In addition, there should be a sequence-structure relationship analysis of the mutations. More importantly, the structure of pyrin molecule should be provided along with the mutated structures.

Authors’ reply :  We added a new figure with the all the mutations found up to now in the MEFV gene and their pathogenicity in section 4 (figure 3). The new figure 1 shows now the structure of the MEFV protein.

  1. Figure 1 caption is not aligned. In the figure, beta is given as “b” and this needs to be changed to the proper symbol. The figure caption for A and B is missing. Different shapes aren’t properly annotated, eg. Square and circle in pyrin molecules.

Authors’ reply : the previous Figure 1 (now Figure 2) was completely modified (see also below). A detailed legend is now provided

  1. Page 2: “ It contains N-terminal PYD domain…” What is the pyd domain? The authors need to explain the pyd domain. The authors need to provide a domain picture of the pyrin protein with their respective functions.

Authors’ reply: We added an explanation for the PYD domain at page 4 in section 3 and we integrated it as suggested with a schematic representation of pyrin protein and its domains (figure 1).

  1. Page 4: “Pyrin interacts with the Yersinia pestis……” The authors need to provide an image of the mechanism explained in the paragraph.  

Authors’ reply As you suggested we also added and image for the mechanisms of Yersinia pestis interaction with pyrin inflammasome assembly (figure 2 C).

  1. It is not clear whether the MEFV gene is dominant or recessive. The authors need to be clear on this part citing proper references.

Authors’ reply: A new paragraph on page 7 better explain the experimental evidences supporting the gain of function consequences of MEFV gene mutations with the related reference.

  1. The mechanism of CRP and SAA levels during the disease needs to be explained in detail with a figure.

Authors’ reply: The elevation of CRP and SAA elevation during disease flare in FMF is rather nonspecific and already described in many reviews. We leave to the Editor the decision to add also this possible figure.

  1. The mechanism of gal-2 and how the same can be a biomarker needs to be explained in detail with a figure.

Authors’ reply: we do apologize with the reviewer. However it was really difficult to identify a study in the literature showing the possible role of gal-2 in FMF. Even the detailed analysis of a recent review on the role of Galectin-2 published in your journal (Galectin-2 in Health and Diseases Int J Mol Sci 2022 Dec 25;24(1):341) did not help us in identify a possible link with FMF. If the Editor think this aspect is crucial for the present review we will be more than happy to implement the text. However we should need some more indications from reviewer 2. indications.

  1. Beta should be given as a symbol in all instances.

Authors’ reply Thank you for the suggestion, we changed beta to the symbol.

  1. Page 7: x-ray should be given as X-ray.

       Authors’ reply Thank you for the suggestion, we have corrected it as indicated.

  1. Page 9: position 242 should be given as single letter residue and the residue number.

        Authors’ reply Thank you for the suggestion, we have corrected it as indicated.              

  1. Page 12: Strength B: What is this?

       Authors’ reply The strength refers to the level of evidence coming from the literature, based on the principles of evidence based medicine.  The strength of a recommendation is related to the quality of the information coming from the literature review.  (Dougados M, Betteridge N, Burmester GR, et alEULAR standardised operating procedures for the elaboration, evaluation, dissemination, and implementation of recommendations endorsed by the EULAR standing committees. Annals of the Rheumatic Diseases 2004;63:1172-1176.). For example, strength A indicates a strong clinical recommendation coming from high quality prospective cohort study with adequate power or systematic review of these studies; strenght B a less strong recommendation coming from lesser quality prospective cohort, retrospective cohort study, or systematic review of these studies. These concepts are largely know by clinicians.

  1. Page 13: Strength A: What is this?

        Authors’ reply : See point 14 above.

  1. Page 17, 18 and 19: The figure numbers are missing and are given as letters. Why?

        Authors’ reply Thank you for the suggestion, we changed it to numbers.

  1. Page 13: “Colchicum Autumnale”…a should be in lower caps.

        Authors’ reply Thank you for the suggestion, we have corrected it as indicated.

  1. Section 9.2: The section is unnecessary and does not add significantly to the review. This portion    should be excluded.

Authors’ reply Thank you for the suggestion. However, since being the review meant to provide to clinicians also complete information novel details on treatment, it should include also treatment side effects.

  1. Section 9.3: “According to the ongoing….” What do the authors mean by 0.5mg/die? What does die mean in the unit?

Authors’ reply Die means “per day”, we changed it in the text.

  1. Figures Z, Y and XX are based on the published data and so proper citation must be provided in the figure captions.

Authors’ reply the proper citation is provided in the figures’ captions.

  1. A future perspective of the review should be given along with concluding remarks.

Authors’ reply: a final paragraph on the future perspective and final remarks is now provided

  1. The binding and mechanism of the IL-1 antibodies should be given with proper structures and interaction data.

        Authors’ reply : the detailed analysis of the structural interaction between IL-1b and anti IL-1 monoclonal is surely of interest, but indeed far from the aim of the review that is mainly aimed to give an update on the pathogenesis, clinical presentation and state of the art on treatment. We leave to the Editor the choice to include this further information in piece of he present review

  1.    There are several grammatical and typographical errors that the authors need to proofread. The representation of scientific data should be uniform and according to the norms.

Authors’ reply Thank you for the suggestion, we have tried to correct al the grammatical and typographical errors in the text.

Round 2

Reviewer 2 Report

1. There are certain characters in figure 1 and 2 which have red underlines. This needs to be corrected. 

Author Response

  1. There are certain characters in figure 1 and 2 which have red underlines. This needs to be corrected.

Authors’ reply: Thank you for the suggestion, we have provided to change the figures according to your suggestion.